rsos.royalsocietypublishing.org

materials science

insulation materials, anti-infrared radiation, composite xerogel

**Authors for correspondence:**
Liu Weijun
e-mail: lwj1119@139.com
Zhang Shuhua
e-mail: zsh_7474@126.com

This article has been edited by the Royal Society of Chemistry, including the commissioning, peer review process and editorial aspects up to the point of acceptance.

# Improving anti-infrared radiation and heat insulation by potassium hexatitanate whisker-doped $Al_2O_3-SiO_2$ composite xerogel

Liang Furong[1], Liu Weijun[1], Han Xiuxiu[2], Zhang Boru[1] and Zhang Shuhua[2]

[1]College of Mechanical and Automobile Engineering, and [2]College of Chemistry and Chemical Engineering, Shanghai University of Engineering Science, Shanghai 201620, People's Republic of China

LF, 0000-0003-2119-6412; LW, 0000-0003-4715-822X

Taking potassium hexatitanate whisker (PTW) modified by silane coupling agent KH550, aluminium nitrate inorganic salt and tetraethyl orthosilicate and deionized water, respectively, as infrared sunscreen, aluminium source and raw materials forming the network structure of a composite xerogel, a series of structurally intact PTW-doped $Al_2O_3-SiO_2$ composite xerogel thermal-insulating materials were prepared by the sol–gel method and dried under atmospheric pressure. The infrared spectral characteristics, infrared radiation transmittance, microstructures, morphology and thermal conductivity of different composite materials prepared have been determined by a Fourier transform infrared spectrometer, UV–visible–near-infrared spectrophotometer, X-ray diffractometer, scanning electron microscope and thermal conductivity tester. The results exhibit that when the Al to Si molar ratio is 1 : 9, the composite material with 5 wt% modified PTW shows the best infrared radiation blocking performance and the lowest thermal conductivity ($0.0604\ W\ m^{-1}\ K^{-1}$).

# 1. Introduction

With the rapid development of the automobile industries, consumers increasingly go for comfortable riding environments. However, we have to sacrifice fuel saving, environment-friendliness and good performance of the vehicle power by using an air conditioner. Therefore, it is necessary for us to study thermal-insulating materials in the era with intense attention to energy and environmental protections. Insulating material is a

rsos.royalsocietypublishing.org   R. Soc. open sci. **5**: 180787

type of porous and loose material with low thermal conductivity which blocks the heat flow transmission. Its thermal conductivity is a combined effect of heat conduction, convective heat transfer and radiant heat transfer. $SiO_2$ xerogel is lightweight and nano-porous with large specific surface area (greater than 800 m$^2$ g$^{-1}$), high porosity (80–99%) and small pore size. The gas-phase conduction and solid-phase conduction of $SiO_2$ xerogel can reach the minimum under the normal temperature and pressure, so it has widely been popular in aerospace, energy, chemical engineering, construction and many other fields [1]. $SiO_2$ xerogel shows poor properties in blocking infrared radiations while most heat of sunlight is concentrated in the infrared radiation band [2]. Therefore, if the infrared radiation can be effectively blocked, the temperature in the vehicle can be lower. In the present study, infrared opacifiers have been doped into a matrix to reduce the infrared radiation permeability. As infrared opacifiers have a great effect on the scatter and absorption of solar radiation, we can greatly increase the extinction coefficient of the xerogel and reduce the radiation thermal conductivity of the high temperature by adding suitable ones. Potassium hexatitanate whisker (PTW), which shows a low thermal conductivity of 5.4 W m$^{-1}$ K$^{-1}$ at room temperature and a lower thermal conductivity value of 1.7 W m$^{-1}$ K$^{-1}$ at a temperature of 800°C, is the main infrared opacifier with good thermal insulation, excellent infrared reflectivity, stable chemical properties, low thermal conductivity and high infrared reflectance and can effectively reduce the transmission of infrared radiation [3].

In general, the thermal conductivity of xerogel is mainly composed of three parts: solid-state heat conduction, gaseous heat conduction and radiant heat conduction [4]. Solid-state heat conduction mainly refers to the thermal conduction of solid particles composing its network structure. Gaseous heat conduction refers to the thermal conduction of gas molecules in nano-pores, consisting of convective heat transfer and heat conduction of gas itself; as the size of pores in the xerogel skeleton is only dozens of nanometres, limiting the thermal movement of the molecule greatly, the convective heat conduction can be neglected which means the gaseous heat conduction is only the gas heat [1].

The three-dimensional network skeleton of the xerogel guarantees its characters of ultra-low density and ultra-low thermal conductivity. The tiny three-dimensional network with pores connected to each other constitutes a complex and tortuous nano-porous structure for heat transferring, making the aperture too small to conduct gas freely. Besides, the addition of the opacifier can greatly improve the radiation resistance of the material. All these provide the material with a low thermal conductivity. In addition, inorganic xerogel is resistant to high temperatures. $SiO_2$ xerogel can withstand 1050°C while 2000°C for $Al_2O_3$ xerogel [5,6]. $Al_2O_3$–$SiO_2$ composite xerogel not only shows advantages of $SiO_2$ xerogel in thermal isolation but also does better in high strength, high thermal and chemical stability so that it overcomes disadvantages of $SiO_2$ xerogel in short usable temperature range, low strength and fragility. In conclusion, $Al_2O_3$–$SiO_2$ composite xerogel shows unparalleled advantages as a high-temperature insulation material [7].

As mentioned above, $Al_2O_3$–$SiO_2$ composite xerogel has attracted much attention for characteristics such as porosity, low density, high strength, stable thermal and chemical properties. As this kind of material can overcome the shortcomings of the $SiO_2$ xerogel of low usable temperature range, low strength and brittleness [5], we have prepared a series of structurally complete $Al_2O_3$–$SiO_2$ composite xerogels by taking the inexpensive aluminium nitrate $(Al(NO_3)_3 \cdot 9H_2O)$ inorganic salts as the aluminium source, compositing the network structure of the composite xerogel with the hydrolysis of organosilicon alkoxides. At the same time, a modified infrared opacifier, PTW, is added to further reduce the heat penetration of infrared radiation.

# 2. Experimental

## 2.1. Main reagents

The main reagents used in the experiment were tetraethyl orthosilicate (analytically pure), anhydrous ethanol, deionized water, aluminium nitrate $(Al(NO_3)_3 \cdot 9H_2O)$, KH-550, $n$-heptane (analytically pure), and PTW (Nanjing Quanxi Chemical Co., Ltd). All reagents above were used directly without further purification. Ethanolic hydrochloric acid solution (0.2 mol l$^{-1}$), 25% ammonia water, 2 mol l$^{-1}$ ethanolic ammonia solution and 70% ethyl orthosilicate ethanol solution were configured by certain proportions.

## 2.2. Methods of characterization and instruments

The surface functional groups for prepared samples were detected by a Fourier transform infrared (FT-IR) instrument (Thermo Fisher Scientific, NICOLET iS10). The crystal structures of the samples were

rsos.royalsocietypublishing.org    R. Soc. open sci. 5: 180787

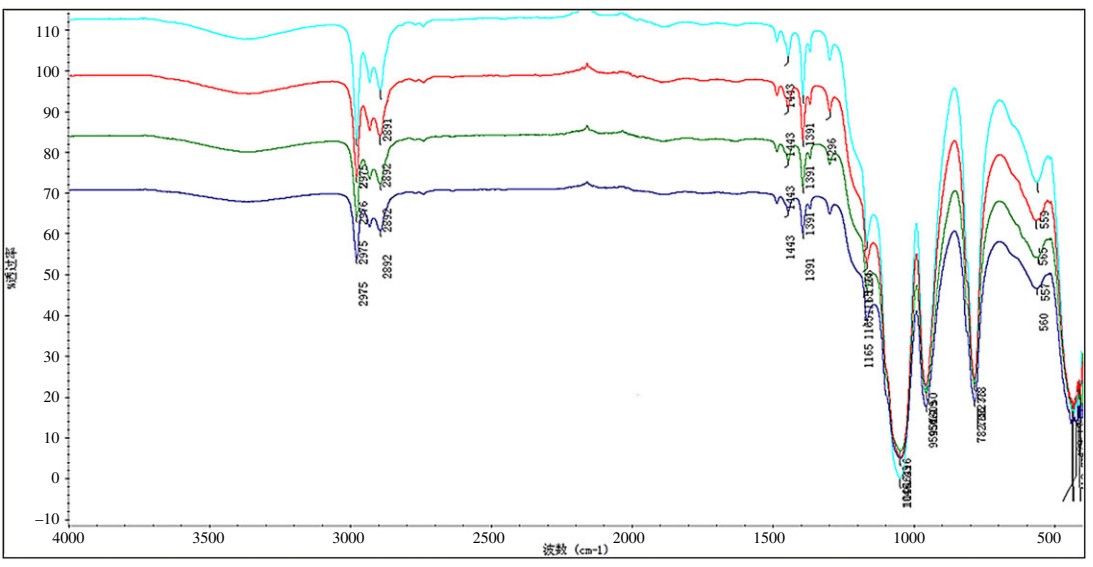

**Figure 1.** Comparison of FT-IR spectra of different modified PTW dosing samples.

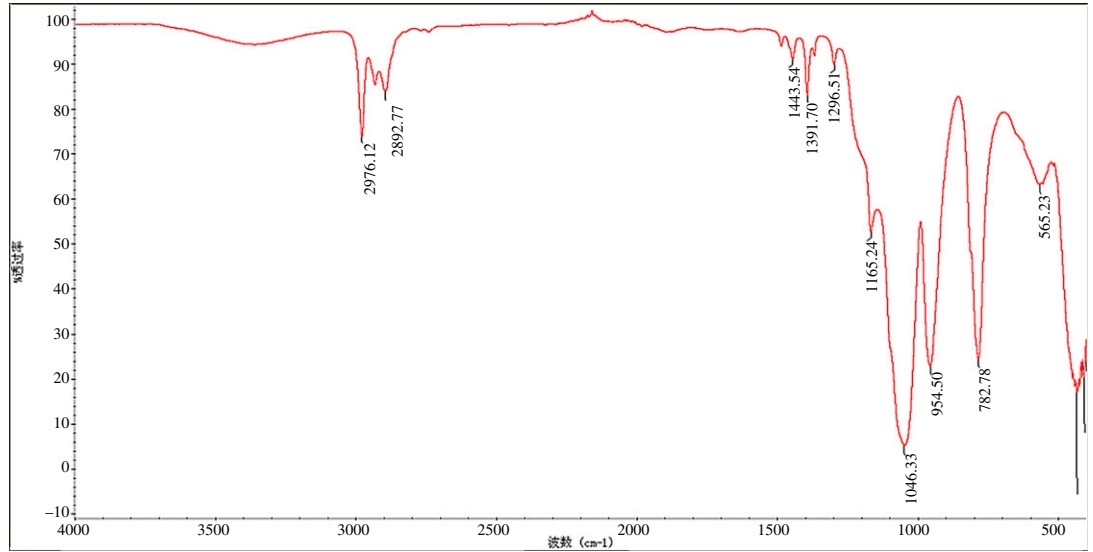

**Figure 2.** FT-IR spectrum of composite xerogel with 5 wt% modified PTW.

determined by X-ray diffraction (XRD; D2 Phaser, Bruker, Germany) at 35 kV and 30 mA. The surface microstructure of the samples was observed by a field-emission scanning electron microscope (model SU8000, Hitachi, Japan). A UV–visible–near-infrared spectrophotometer (UV-3600) was used to record the infrared radiation barrier properties of the samples. The thermal conductivity of the samples with different proportions was tested by DRE-III thermal conductivity tester (Xiangtan Xiangyi Instrument Co., Ltd). Ultrasonic dispersion analyser (Shanghai Branch Ultrasonic Instrument Co. Ltd, SK7210HP), vacuum drying chamber (DZF-6021, Shanghai Yiheng Scientific Instrument Co. Ltd), DF-101S heat-collecting magnetic heating stirrer (Jintan Ronghua Instrument Manufacturing Co. Ltd), and high-temperature box type muffle furnace (Shanghai Shinji Co. Ltd, SG-XS1200) were also used for the preparations in the experiment.

## 2.3. Preparation and characterization

### 2.3.1. Modification of potassium hexatitanate whisker

After placing 10 g of PTW into 100 ml anhydrous ethanol, 0.5 ml KH-550 was added and magnetically stirred for 1 h before drying at 50°C to obtain modified PTW.

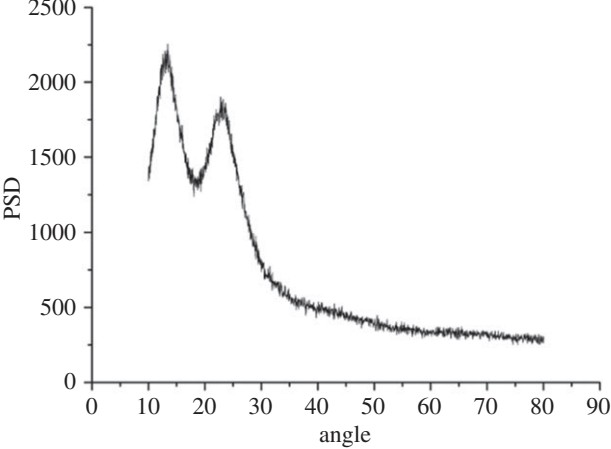

**Figure 3.** XRD pattern of SiO$_2$ xerogel.

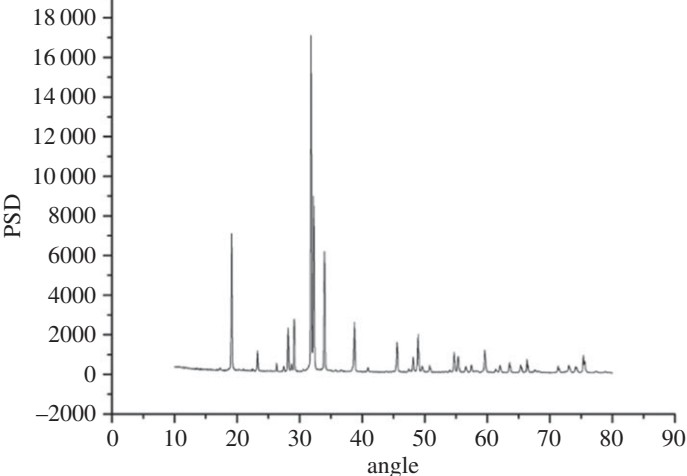

**Figure 4.** XRD pattern of modified PTW.

### 2.3.2. Preparation of γ-Al$_2$O$_3$

After mixing Al(NO$_3$)$_3 \cdot$ 9H$_2$O into a 1 mol l$^{-1}$ solution of Al(NO$_3$)$_3$, we stirred it magnetically at 60°C before 25% aqueous ammonia solution was added dropwise until the pH of the solution reached 9.0– 9.5. A stable white precipitate (γ-AlOOH) formed after finishing every step above. Then we filtered the white solid and washed it for 3–4 times before drying it in a constant temperature oven at 50°C for a period of time. After that, dry white precipitates were placed in a muffle furnace and heated up slowly to 450°C for 2–3 h before cooling to ambient temperature with furnace. After finishing all steps listed, we ground the final white precipitate to ultrafine powders.

## 2.4. Preparation of new thermal-insulating and radiation-blocking Al$_2$O$_3$ – SiO$_2$ composite xerogel doped with modified potassium hexatitanate whisker

After adding a certain amount of the modified PTW and γ-Al$_2$O$_3$ to anhydrous ethanol and dispersing ultrasonically and uniformly while adding tetraethyl orthosilicate, deionized water and anhydrous ethanol in the molar ratio of 1 : 4 : 10, an appropriate amount of 0.2 mol l$^{-1}$ ethanolic hydrochloric acid and 2 mol l$^{-1}$ ethanolic ammonium hydroxide solution, which were used to adjust the reaction pH, were added at 50°C and stirred magnetically. The gel appeared in several minutes. After being dried in a thermostat at 50°C for a period of time, the gel was soaked in 70% ethyl orthosilicate ethanol solution in a vacuum drying oven at 50°C for 24 h; then the gel was added to *n*-heptane in a vacuum

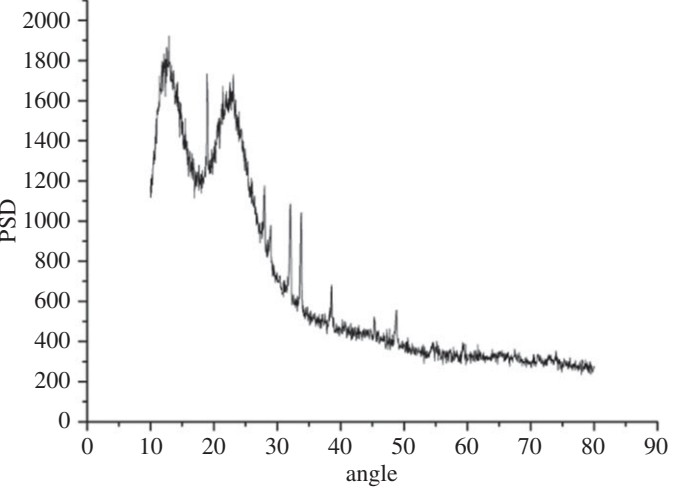

**Figure 5.** XRD pattern of composite $Al_2O_3 - SiO_2$ xerogel with 5 wt% modified PTW.

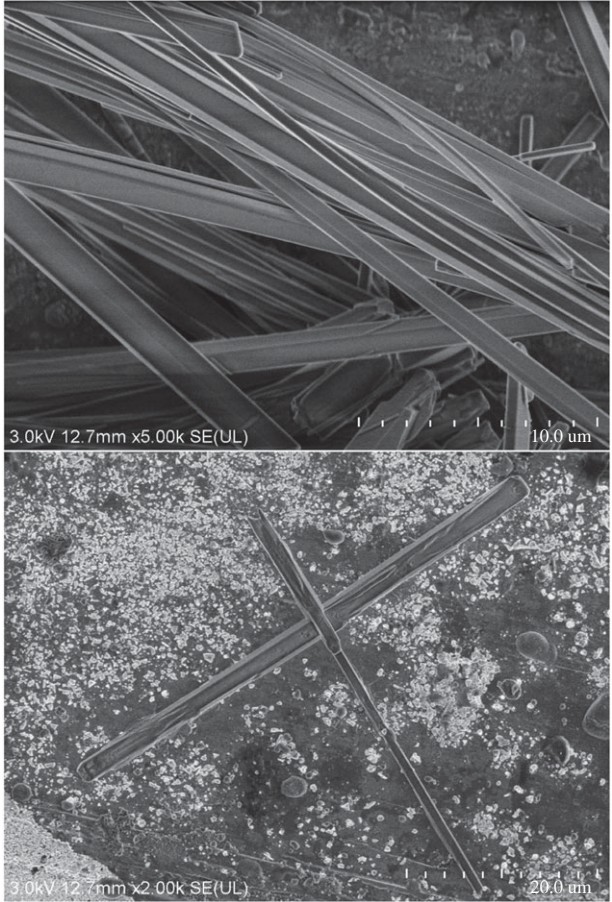

**Figure 6.** Electron microscope images of PTW.

drying oven at 50°C as the solvent replacements for 2–3 times. Finally, we dried the gel at 50°C for almost one week to transform it into the xerogel.

## 3. Results and discussion

To facilitate the comparative analysis, a set of samples were prepared in experiments: $SiO_2$ xerogel, composite materials doped with modified PTW at 0 wt%, 5 wt%, 10 wt% and 15 wt% with molar ratio

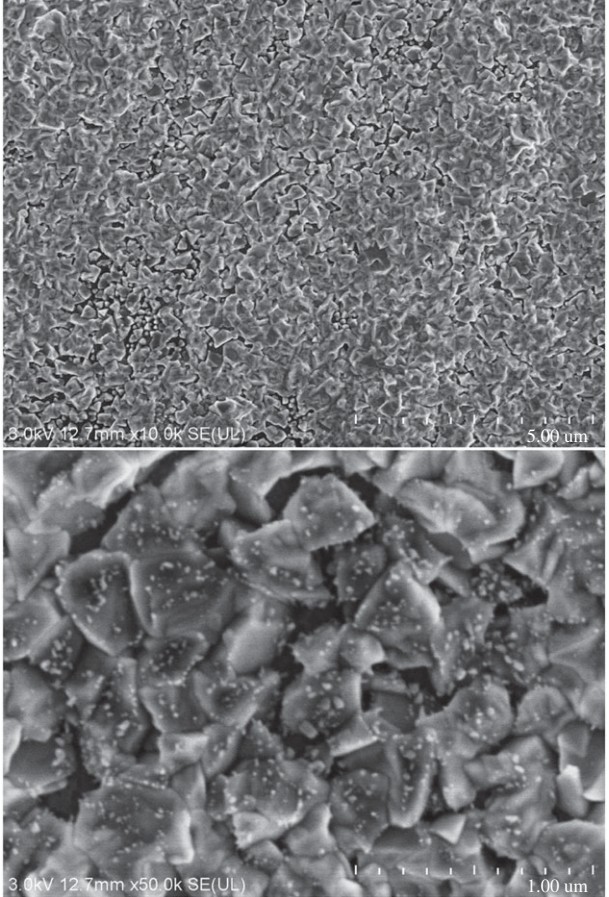

**Figure 7.** Electron microscope images of sample II.

of Al and Si as 1 : 9, and composite materials doped with modified PTW at 5 wt% with molar ratio of Al and Si, respectively, as 2 : 8 and 3 : 7 which were numbered as I, II, III, IV, V, VI, VII as shown below. For convenience and clarity, we take the sample III as an example for the specific analysis of the new composite materials.

## 3.1. Analysis of microstructure of new-prepared composite materials

### 3.1.1. Infrared analysis of new composite materials

The infrared contrastive spectra (figure 1) of four samples with different amount of modified PTW at room temperature, II, III, IV and V, reveal basically the same peak positions of four samples, which indicated that the different doping amount of PTW brought no difference to main compositions of the final products prepared. Here, we specially put all curves of these four samples in figure 1 to clearly compare those peak positions.

To specifically analyse the structures and functional groups of the new composite materials, the FT-IR spectrum (figure 2) of the composite xerogel with a modified PTW doping amount of 5 wt% is taken out individually as an example. It exhibits that the absorption peak at 2976.12–2892.77 $cm^{-1}$ is ascribed to the C—H vibration in KH-550, while the absorption peak at 1443.54–1165.24 $cm^{-1}$ is caused by the vibration of organic impurities. In addition, the absorption peak at 1046.33 $cm^{-1}$ corresponds to the stretching vibration of Si—O—Si in xerogel and the absorption peak at 954.50 $cm^{-1}$ is due to the vibration of Si—OH. Meanwhile, the absorption peak at 782.78 $cm^{-1}$ corresponds to the vibration of Si—O—C generated on the surface of $SiO_2$ and KH-550. In the low-frequency region, the absorption peak near 565.23 $cm^{-1}$ results from the vibration of Al—O bond in the γ-$Al_2O_3$. The wavenumber is slightly red-shifted according to the figure. The literature research works have suggested that there are Si—O—Al bonds in the composite oxides, and the Si—OH vibration corresponding to the absorption peak at

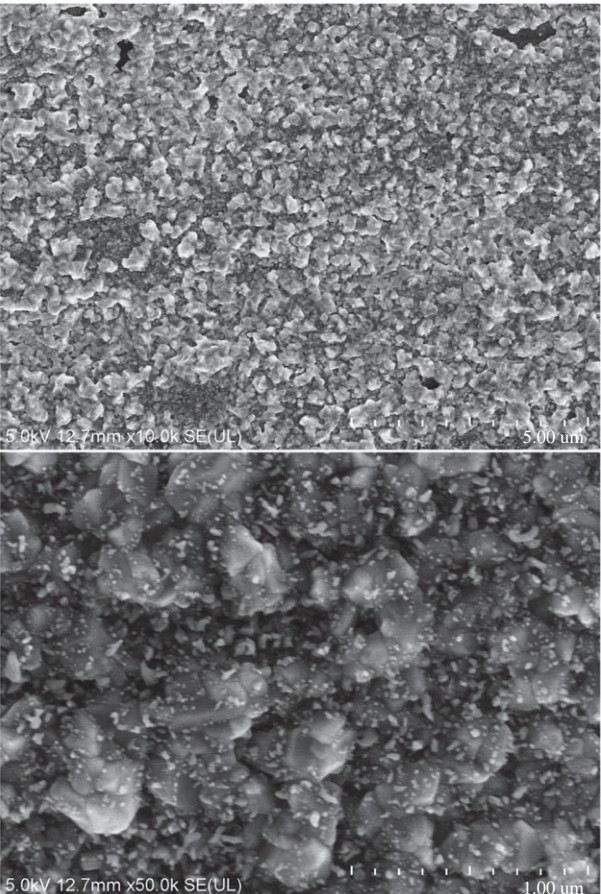

**Figure 8.** Electron microscope images of sample I.

954.50 cm$^{-1}$ is an evidence of the gel homogeneity of Al$_2$O$_3$–SiO$_2$ composites, that is to say, the strength of this band will be reduced with gradual increase of Si—O—Al bond.

### 3.1.2. X-ray diffraction analysis on crystal structures of new-prepared composites

XRD patterns of SiO$_2$ xerogel, modified PTW and Al$_2$O$_3$–SiO$_2$ composite xerogel with modified PTW doping amount of 5 wt% are shown in figures 3–5, respectively. Figure 3 shows humps at 15 degrees and 22 degrees that indicate that SiO$_2$ xerogel prepared is amorphous- and mesoporous-ordered. From figure 4, it can be seen that characteristic peaks of the PTW are obvious indicating that the modified PTW with the addition of KH-550 still keeps a good crystal structure. However, there exist some impurity peaks which may be caused by small amount of potassium tetratitanate produced unexpectedly in the experiment. There exist some new sharp peaks around 18, 34, 38 and 50 degrees shown in figure 5 compared with figure 3. The presence of these new peaks identifies the modified PTW and testifies that the SiO$_2$ xerogel dried under atmospheric pressure incorporating the Al$_2$O$_3$– SiO$_2$ composite xerogel and 5 wt% modified PTW still keeps the amorphous network structure. Besides, the PTW and γ-Al$_2$O$_3$ are well dispersed in the xerogel with no crystalline precipitation.

### 3.1.3. Electron microscopic analysis for surface microstructures of new composites

As we know, electron microscope images can directly reflect the void distribution and particle size of materials. Figures 6–11 show particles' surface morphologies and microstructures of modified PTW and samples I, II, III, IV and V determined by a scanning electron microscope, respectively. As can be seen from the images, xerogel is composed of nano–micro particles. According to figures 7 and 8, the partial aggregation of the particles becomes more apparent by adding Al$_2$O$_3$, and the voids between the agglomerated particles are also reduced. Besides, silica xerogel exhibits certain shrinkage on the

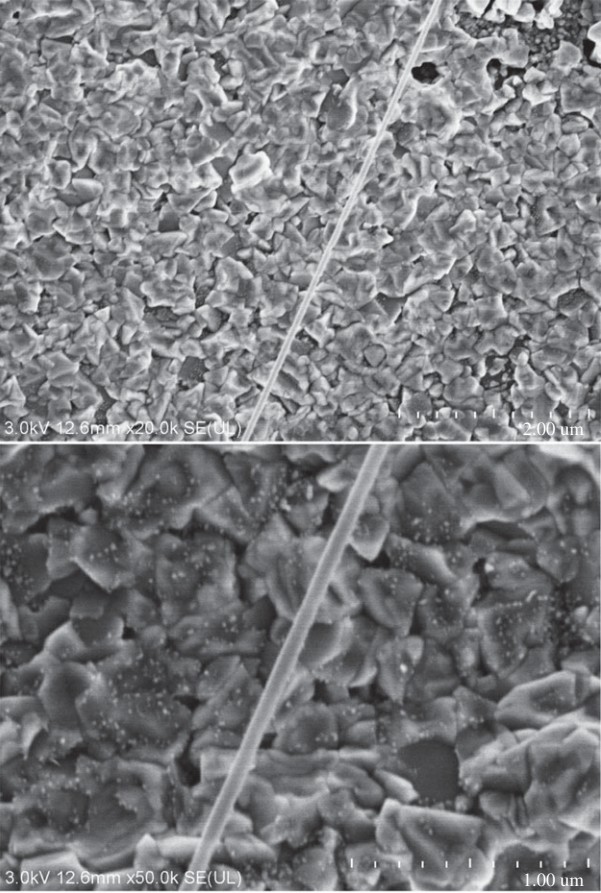

**Figure 9.** Electron microscope images of sample III.

surface due to the preparation method and conditions. After adding $Al_2O_3$, we can see from figure 8 that the shrinkage of sample II has reduced compared with figure 7. When the PTW is added in not too great an amount, according to figure 9, the particle size of the agglomerated particles gets bigger and the voids between the agglomerated particles are further reduced, the shrinkage is further reduced and the structure of sample III is relatively dense and the whiskers are faintly visible. While with the increasing doping amount of PTW, according to figures 10 and 11, what can be seen clearly is that with the increasing amount of PTW in the xerogel, the size of the agglomerated particles gradually becomes larger and the compactness of the samples is seriously damaged. The structures and related properties of samples are influenced and changed due to the uneven size of PTW particles which can be seen as long strips in figures 9–11 that are much larger than silica particles. (According to figures 6 and 7, the PTW is small, 0.5–1.5 $\mu$m in diameter and 30–50 $\mu$m in length. Although it is not possible to accurately identify the particle size of $SiO_2$ from figure 7, it can be determined that they are nanometre-sized and definitely much smaller than the former.)

## 3.2. Analysis of thermal insulation of new composite materials

### 3.2.1. Thermal conductivity analysis

The corresponding thermal conductivities of experimentally prepared samples are shown in table 1. As can be seen from table 1, the thermal conductivity of the $Al_2O_3$–$SiO_2$ composite xerogel is slightly higher than that of the $SiO_2$ xerogel. It is because the relative solid-state heat conduction of the $Al_2O_3$–$SiO_2$ composite xerogel is more due to the lower porosity by adding $\gamma$-$Al_2O_3$ to the network structure than $SiO_2$ xerogel. With the amount of PTW increasing, the thermal conductivity of the composite xerogel decreases and then increases. This is because when PTW is not over-incorporated, with the increasing incorporation of PTW, the total thermal conductivity of the composite xerogel decreases owing to the decrease of the radiant thermal conductivity. But when a certain amount of PTW is added, the excess

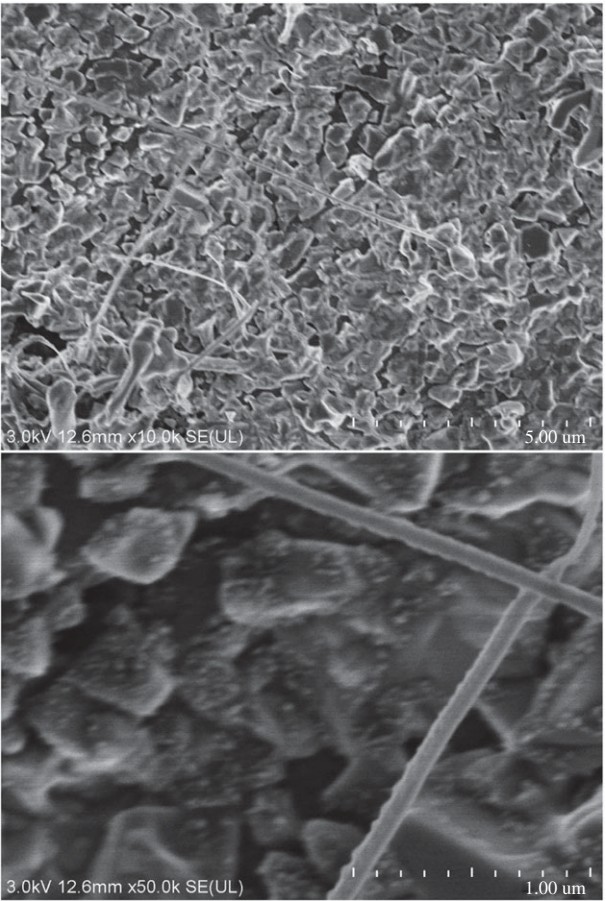

**Figure 10.** Electron microscope images of sample IV.

powder may cover the colloidal particles, blocking the openings, resulting in a decrease in the specific surface area then affecting the normal structure of the xerogel which leads to a greater increase in solid-state thermal conductivity and less of a decrease in the radiative thermal conductivity. Besides, according to the comparison of samples III, VII and VIII which shows a trend that the thermal-insulating performance of the samples decreases as the amount of Al increases, we decided to choose Al to Si molar ratio of 1 : 9 to prepare the composite xerogel in the experiment.

It leads to an increase in overall thermal conductivity of the composite xerogel. This uptrend is consistent with that in infrared thermal radiation blocking performance with the incorporation of PTW. The lowest thermal conductivity in table 1 is $0.0604 \, \mathrm{W \, m^{-1} \, K^{-1}}$ and the PTW incorporation is 5 wt%, which should be the most appropriate PTW incorporation in this study.

### 3.2.2. Infrared radiation blocking performance analysis

Figure 12 shows the graph of infrared transmittance of samples I, II, III, IV and V in the 400–1400 nm band measured at the normal temperature.

The concentration and thickness of samples are kept the same during tests. It can be seen from the graph that $Al_2O_3$–$SiO_2$ composite xerogel shows a lower infrared transmittance in the 400–1400 nm band compared with $SiO_2$ xerogel because of the low thermal conductivity as well as the good infrared blocking properties of the $Al_2O_3$ in the composite xerogel [5]. It provides the composite xerogel with smaller thermal conductivity and better infrared radiation blocking performance than $SiO_2$ xerogel [8]. Meanwhile, the infrared transmittance of the $Al_2O_3$–$SiO_2$ composite xerogel doped with 5 wt% of modified PTW is the lowest because of the presence of PTW which is a high-infrared-reflective material. Doped with an appropriate amount of PTW, the composite xerogel pore structure will be more uniform. That is to say, when the sunlight passes through the xerogel doped with PTW, it will reflect a part of the infrared light to avoid the absorption of infrared light caused by its own rising temperature [9]. PTW particles can reflect and absorb infrared light inside the gel and increase

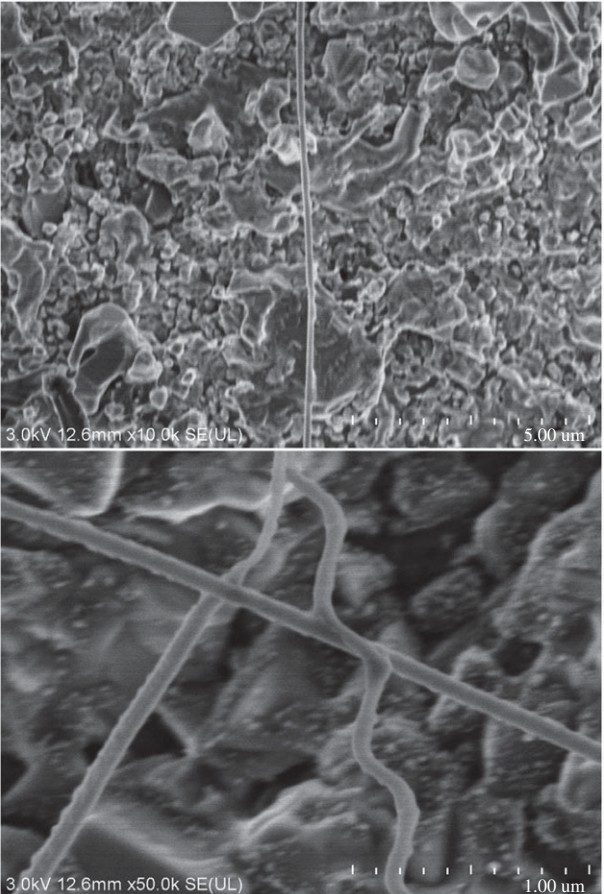

**Figure 11.** Electron microscope images of sample V.

**Table 1.** Different modified PTW ratios corresponding to the thermal conductivity of the composite xerogel.

| sample | content of Al (%) | content of PTW (%) | volumetric heat capacity (MJ m$^{-3}$ K$^{-1}$) | thermal diffusivity (mm$^2$ s$^{-1}$) | thermal conductivity (W m$^{-1}$ K$^{-1}$) |
|---|---|---|---|---|---|
| I | 0 | 0 | 0.3199 | 0.20 | 0.0640 |
| II | 10 | 0 | 0.2338 | 0.32 | 0.0748 |
| III | 10 | 5 | 0.3353 | 0.18 | 0.0604 |
| IV | 10 | 10 | 0.5701 | 0.14 | 0.0658 |
| V | 10 | 15 | 0.4663 | 0.15 | 0.0699 |
| VI | 20 | 5 | 0.4369 | 0.14 | 0.0612 |
| VII | 30 | 5 | 0.4399 | 0.12 | 0.0639 |

light attenuation. However, with the further increasing doping amount of the modified PTW, the infrared radiation blocking ability is gradually weakened. The reason is that the incorporation of PTW is equivalent to that of defects in the composite xerogel. As the amount of PTW doping increases, the size of the agglomerated particles in the samples gradually becomes larger and the compactness is seriously damaged which probably leads to the detrimental effect on the normal structures and related properties of samples because the uneven PTW particles are much larger than silica particles. Therefore, when doping too much PTW, the infrared radiation blocking ability of the overall composite xerogel may be adversely affected. In conclusion, considering the properties of thermal conductivity analysis of this new type of insulated composite xerogel, the thermal insulation performance will be best when the amount of PTW incorporated is 5 wt%.

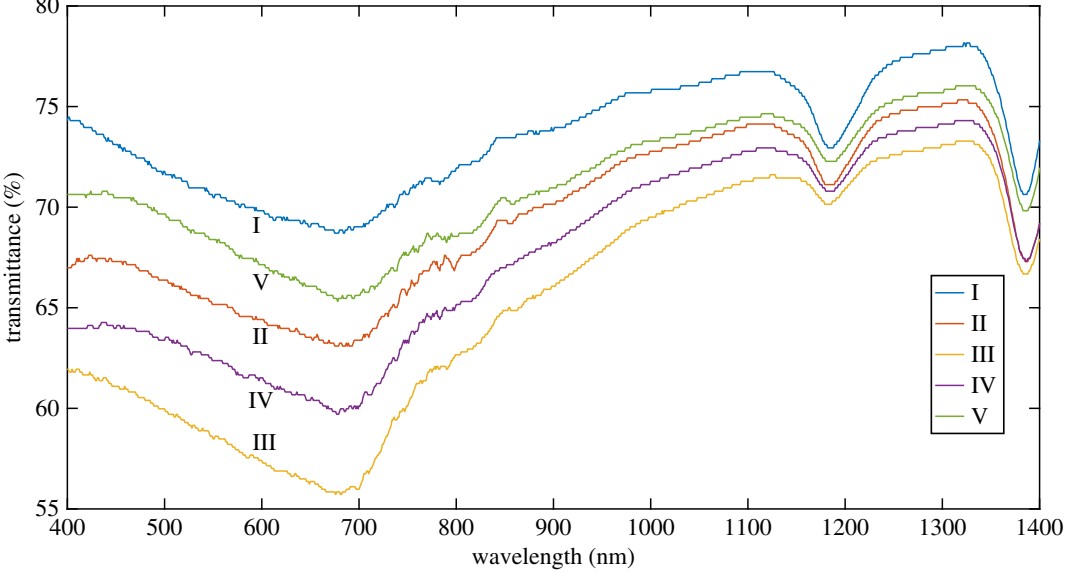

**Figure 12.** Infrared transmittance in the 400–1400 nm band of composite xerogel with different PTW doping levels.

## 4. Conclusion

PTW surface-modified with KH-550 disperses uniformly in $Al_2O_3$–$SiO_2$ composite xerogel and provides new thermally insulated composite xerogel powder doped with infrared sunscreen with better completed material structure.

When the amount of PTW added is appropriate, it penetrates the network structure of the composite xerogel and maintains the original crystal structure. At the same time, the new composite xerogel prepared can also maintain the original amorphous network structure. It shows that the PTW and γ-$Al_2O_3$ can be well dispersed in the xerogel without precipitating.

The infrared transmittances in the 400–1400 nm range measured at room temperature show that the infrared light transmittance ratio of the $Al_2O_3$–$SiO_2$ composite xerogel is lower than that of the $SiO_2$ xerogel. According to the experimental data, when the molar ratio of Al and Si is 1:9, the thermal conductivity of the sample with 5 wt% of modified PTW reaches the minimum (0.0604 W m$^{-1}$ K$^{-1}$).

Considering both the infrared radiation blocking property and thermal conductivity of the new thermal-insulated and infrared composite xerogel prepared, when the molar ratio of Al and Si is 1:9, the sample with 5 wt% of modified PTW shows the best performance.

Data accessibility. All the available data for this work are presented within the paper.

Authors' contributions. L.W. and Z.S. designed the study. L.F. prepared all samples for analysis, interpreted the results and wrote the manuscript. All authors gave final approval for publication.

Competing interests. We declare we have no competing interests.

Funding. Financial support came from University Students' Research and Innovation Projects of China and Shanghai (17KY0114).

Acknowledgements. We thank College of Chemistry and Chemical Engineering and College of Mechanical Engineering, Shanghai University of Engineering Science for providing laboratory equipment and testing equipment. We are also grateful to Xiuxiu Han and Boru Zhang for their assistance in using equipment and providing the chemical reagents.

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
