## [Reviewer comments · Royal Society Open Science]

Review History

RSOS-180787.R0 (Original submission)

Review form: Reviewer 1

Is the manuscript scientifically sound in its present form?

Yes

Are the interpretations and conclusions justified by the results?

Yes

Is the language acceptable?

Yes

Is it clear how to access all supporting data?

Not Applicable

Do you have any ethical concerns with this paper?

No

Have you any concerns about statistical analyses in this paper?

I do not feel qualified to assess the statistics

Recommendation?

Major revision is needed (please make suggestions in comments)

Comments to the Author(s)

The authors improved Anti-infrared Radiation and Heat-insulation of composite aerogel by adding PTW. The work is interesting and can be publish in this journal if the following issues are addressed:

1. In the Thermal conductivity analysis section, Paragraph 1 and 2 should be moved to the Introduction. These paragraphs review on thermal conductivity of material.
2. Why choose the molar ration of Al and Si is 1:9.
3. What difference between FI-IR results of modified PTW dosing samples and PTW doped composite aerogel (Figure 1 and 2).
4. What about xerogel particle size? The SEM results did not clearly show the difference of xerogel particle size as authors mentioned. The authors should add SEM results with higher magnification to prove the discussion.
5. Shrinkage of gel can be prove by porosity of samples. Porosity can be calculate by density. The authors should add these results and combine with SEM to prove your discussion on structure of composite aerogel.
6. How to compare particle size of PTW and SiO₂ with no data. Should add these data to clarify.
7. Why adding PTW lead to reduce thermal conductivity of composite aerogel. What is the thermal conductivity value of PTW.
8. English writing need to be improved.

Review form: Reviewer 2

Is the manuscript scientifically sound in its present form?

Yes

Are the interpretations and conclusions justified by the results?

Yes

Is the language acceptable?

Yes

Is it clear how to access all supporting data?

Yes

Do you have any ethical concerns with this paper?

No

Have you any concerns about statistical analyses in this paper?

No

Recommendation?

Accept as is

Comments to the Author(s)

The paper "Improving Anti-infrared Radiation"

is interesting and actual. I may recommend this paper for publication.

I have not any importante comments or remarks.

Decision letter (RSOS-180787.R0)

03-Oct-2018

Dear Miss Liang:

Title: Improving Anti-infrared Radiation and Heat-insulation by PTW-doped Al₂O₃-SiO₂ Composite Xerogel
Manuscript ID: RSOS-180787

The editor assigned to your manuscript has now received comments from reviewers. We would like you to revise your paper in accordance with the referee and Subject Editor suggestions which can be found below (not including confidential reports to the Editor). Please note this decision does not guarantee eventual acceptance.

Please submit your revised paper before 26-Oct-2018. Please note that the revision deadline will expire at 00.00am on this date. If we do not hear from you within this time then it will be assumed that the paper has been withdrawn. In exceptional circumstances, extensions may be possible if agreed with the Editorial Office in advance. We do not allow multiple rounds of revision so we urge you to make every effort to fully address all of the comments at this stage. If deemed necessary by the Editors, your manuscript will be sent back to one or more of the original reviewers for assessment. If the original reviewers are not available we may invite new reviewers.

Once again, thank you for submitting your manuscript to Royal Society Open Science and I look

forward to receiving your revision. If you have any questions at all, please do not hesitate to get in touch.

Yours sincerely,
Dr Laura Smith, MRSC
Publishing Editor, Journals
Royal Society of Chemistry,
Thomas Graham House,
Science Park, Milton Road,
Cambridge, CB4 0WF, UK

Royal Society Open Science - Chemistry Editorial Office

On behalf of the Subject Editor Professor Anthony Stace and the Associate Editor Professor Claire Carmalt.

RSC Associate Editor:
Comments to the Author:
(There are no comments.)

RSC Subject Editor:
Comments to the Author:
(There are no comments.)

Reviewers' Comments to Author:
Reviewer: 1

Comments to the Author(s)

The authors improved Anti-infrared Radiation and Heat-insulation of composite aerogel by adding PTW. The work is interesting and can be publish in this journal if the following issues are addressed:

1. In the Thermal conductivity analysis section, Paragraph 1 and 2 should be moved to the Introduction. These paragraphs review on thermal conductivity of material.
2. Why choose the molar ration of Al and Si is 1:9.
3. What difference between FI-IR results of modified PTW dosing samples and PTW doped composite aerogel (Figure 1 and 2).
4. What about xerogel particle size? The SEM results did not clearly show the difference of xerogel particle size as authors mentioned. The authors should add SEM results with higher magnification to prove the discussion.
5. Shrinkage of gel can be prove by porosity of samples. Porosity can be calculate by density. The authors should add these results and combine with SEM to prove your discussion on structure of composite aerogel.
6. How to compare particle size of PTW and SiO₂ with no data. Should add these data to clarify.
7. Why adding PTW lead to reduce thermal conductivity of composite aerogel. What is the thermal conductivity value of PTW.
8. English writing need to be improved.

Reviewer: 2

Comments to the Author(s)

The paper "Improving Anti-infrared Radiation"

is interesting and actual. I may recommend this paper for publication.

I have not any importante comments or remarks.

Author's Response to Decision Letter for (RSOS-180787.R0)

See Appendix A.

RSOS-180787.R1 (Revision)

Review form: Reviewer 1

Is the manuscript scientifically sound in its present form?

Yes

Are the interpretations and conclusions justified by the results?

Yes

Is the language acceptable?

Yes

Is it clear how to access all supporting data?

Yes

Do you have any ethical concerns with this paper?

No

Have you any concerns about statistical analyses in this paper?

No

Recommendation?

Accept as is

Comments to the Author(s)

My comments are addressed well. I suggest to accept it. Many thanks.

Decision letter (RSOS-180787.R1)

06-Nov-2018

Dear Miss Liang:

Title: Improving Anti-infrared Radiation and Heat-insulation by PTW-doped Al₂O₃-SiO₂ Composite Xerogel
Manuscript ID: RSOS-180787.R1

It is a pleasure to accept your manuscript in its current form for publication in Royal Society Open Science. The chemistry content of Royal Society Open Science is published in collaboration with the Royal Society of Chemistry.

On behalf of the Subject Editor Professor Anthony Stace and the Associate Editor Professor Claire Carmalt.

RSC Associate Editor:
Comments to the Author:
(There are no comments.)

RSC Subject Editor:
Comments to the Author:
(There are no comments.)

Reviewer(s)' Comments to Author:
Reviewer: 1

Comments to the Author(s)
My comments are addressed well. I suggest to accept it. Many thanks.

Appendix A

Dear referees:

First of all, please accept my apology for giving you a reply after such a long time. Due to the occupancy of the test equipment, the re-test of the sample's SEM results has delayed some time.

It is my honor that you can read this article patiently during your busy schedule and I am really appreciated for your recognition of this article. Besides, thank you for providing these instructive suggestions which do a lot of help to the further improvement of the article. I have carefully considered the issues you addressed and have revised them one by one in this article. The specific modifications are shown in Annex I on the following pages.

Thank you again for your suggestions. I hope the answers and modifications I give are valid and appropriate. If there is any other issue or suggestion, please do not hesitate to let me know and I am willing to provide the further modifications.

Annex I:

Manuscript ID: RSOS-180787

Title: Improving Anti-infrared Radiation and Heat-insulation by PTW-doped Al₂O₃-SiO₂ Composite Xerogel

Point-by-Point Responses:

Reviewer 1

Comments: The authors improved Anti-infrared Radiation and Heat-insulation of composite aerogel by adding PTW. The work is interesting and can be published in this journal if the following issues are addressed.

Comments:

1. In the Thermal conductivity analysis section, Paragraph 1 and 2 should be moved to the Introduction. These paragraphs review on thermal conductivity of material.

Respond: After carefully considering your suggestion here, I agree that it is more reasonable to put the first and second paragraph of the Thermal conductivity analysis section into the Introduction. (For the detailed revised content, please check on **page 1-2** of the paper which I have highlighted with the **red font**.)

2. Why choose the molar ratio of Al and Si is 1:9.

Respond: I am sorry for not clearly stating the reasons for this molar ratio in the paper because of the wrong consideration about the conciseness, here we accept your comment to supplement the reason for choosing this ratio in our paper. And for the better logicity, we change the properly the expounding order of thermal conductivity analysis and infrared radiation blocking performance analysis. (For the detailed revised content, please check on **page 4, 5 and 7** of this paper which I have highlighted with the **red font**.) For your convenience, here I have summarized the reason below:

In order to investigate what difference in its thermal conductivity will be made by adding different amount of Al₂O₃ into the composite gel, two other samples adding the Al to Si molar ratio as 2:8 and 3:7 with 5wt% modified PTW have been prepared in the laboratory (sample VI and VII), and their thermal conductivities were measured after drying at 100 °C for one hour (Same processing conditions as sample III), which is higher compared with the experimental sample III.

Sample	Al:Si	Content of PTW/(%)	Volumetric heat capacity (MJ/(m ³ .K))	Thermal diffusivity(mm ² /s)	Thermal Conductivity(W/(m.K))
III	1:9	5	0.3353	0.18	0.0604
VI	2:8	5	0.4369	0.14	0.0612
VII	3:7	5	0.4399	0.12	0.0639

Considering that the increasing amount of Al increases the thermal conductivity instead of improving the thermal insulating property of the samples, we choose the most appropriate Al to Si molar ratio as 1:9 with the lowest thermal conductivity.

3. What difference between FI-IR results of modified PTW dosing samples and PTW doped composite aerogel (Figure 1 and 2).

Respond: I am really sorry that our unclearly expression in this section has caused such doubts. Here I further explain: Fig.1 and Fig.2 are showed in this paper not for comparison. They both show the samples in the infrared spectrum to analyze functional group structures. (For the detailed revised content, please check on **page 4, Paragraph 1 and 2 in the section “Infrared analysis of new composite materials”** of this paper which I have highlighted with the **red font**.) For convenience, here I have summarized the reasons why we present these two figures with no particular differences below:

In order to intuitively compare infrared contrastive spectrums of these four samples with different amount of modified PTW at room temperature: II, III, IV, and V, we put all curves on one image (Fig.1) to clearly reveal that the four samples show the same peak positions, which indicates that the different doping amount of PTW brought no difference to main compositions of the final products prepared. And we take the sample prepared with a modified PTW doping amount of 5wt% (sample III) as an example individually on Fig.2 to specifically analyze the structures and functional groups of the new composite material.

4. What about xerogel particle size? The SEM results did not clearly show the difference of xerogel particle size as authors mentioned. The authors should add SEM results with higher magnification to prove the discussion.

Respond: I am pretty sorry for not providing the suitable SEM results and giving an unclear description and discussion about the products of experimental research due to the poor testing techniques and the inappropriate selection of the SEM results. We have offered more appropriate SEM results and discussions in the revised paper after re-testing the samples. The highest SEM results provided in the revision are magnified by 50.0K times. (For the detailed revised content, please check on **page 4-5, in the section “Electron microscopic analysis for surface microstructures of new composites”** of this paper which we have highlighted with the **red font.**) The specific amendments and SEM results are offered as follows:

First of all, it can be seen from the images that the particle size of the experimentally-prepared xerogel samples are nanometer-scale which means the xerogel is composed by nano-micro particles. According to Fig. 1 and Fig. 2, the partial aggregation of the particles becomes more apparent and the particle size of the agglomerated particles becomes larger by adding Al_2O_3 , and the voids between the agglomerated particles are also reduced. When the adding PTW is not too much, according to Fig. 3, the particle size of the agglomerated particles is getting bigger and the voids between the agglomerated particles are further reduced. While with the increased doping amount of PTW, according to Fig. 4 and Fig. 5, what can be seen clearly is that though the particle size is getting bigger, the voids become larger which could indicate that the compactness of the samples is seriously damaged.

Fig.1 Electron microscope image of sample I

Fig.2 Electron microscope image of sample II

Fig.3 Electron microscope image of sample III

Fig.4 Electron microscope image of sample IV

Fig.5 Electron microscope image of sample V

5. Shrinkage of gel can be proved by porosity of samples. Porosity can be calculate by density. The authors should add these results and combine with SEM to prove your discussion on structure of composite aerogel.

Respond: Thank you for your suggestion here and it is necessary for us to apologize that we realized the improper analysis about the porosity previously due to improper selection and misunderstanding of the SEM results after re-testing the microstructure of the samples. We have put the clearer results and revised the discussion in this section about microscopic structures of xerogel we prepared. (For the detailed revised content, please check on **page 4, 5 and 7** of this paper which we have highlighted in the **red font**.)The specific amendments are summarized as follows:

As can be seen obviously from Fig.1, silica xerogel exhibits certain shrinkage on the surface due to the preparing method and conditions. After adding Al_2O_3 , we can see from Fig.2 that the shrinkage of sample II has reduced comparing with Fig.1. As Fig.3 shows, the shrinkage is further reduced and the structure of the sample III is relatively dense and the whiskers are faintly visible. According to Fig.4 and 5, with the increasing amount of PTW in the xerogel, the compactness of the samples is seriously damaged which finally affect the related properties of samples. The reason probably is that the size of PTW particles which can be seen as long strips in Fig.3,4 and 5 is uneven and much larger than that of the silica particles .

Fig.1 Electron microscope image of sample I

Fig.2 Electron microscope image of sample II

Fig.3 Electron microscope image of sample III

Fig.4 Electron microscope image of sample IV

Fig.5 Electron microscope image of sample V

6. How to compare particle size of PTW and SiO₂ with no data. Should add these data to clarify.

Respond: Following your comment, we have added the particular images for the particle size of PTW and SiO₂ in the revision to clarify the comparison. (For the detailed revised content, please check on **the top of page 5** of this paper which I have highlighted in the **red font**.) For convenience, here I have summarized corresponding analysis below:

According Fig.6, the PTW prepared is small, 0.5-1.5 μ m in diameter and 30-50 μ m in length. As can be seen from Fig.7, the particle size of SiO₂ is nanometer-sized which is much smaller than the former.

Fig.6 Electron microscope images of PTW

Fig.7 Electron microscope image of SiO₂

7. Why adding PTW lead to reduce thermal conductivity of composite aerogel. What is the thermal conductivity value of PTW.

Respond: Thank you for your suggestion here. After a careful consideration, we have made a supplement in the paper about the thermal conductivity value of

PTW and its thermal conductivity is 5.4 W/(m.K) at room temperature. (For the detailed revised content, please check **paragraph 1 on page 1** of this paper which I have highlighted in the **red font**.)

Here is the reason for the reduction on thermal conductivity of composite aerogel by adding PTW: Due to the high infrared reflectivity of PTW, with the appropriate increasing incorporation of PTW, the total thermal conductivity of the composite xerogel decreases owing to the decrease of the radiant thermal conductivity which is benefit from PTW.

8. English writing need to be improved.

Respond: Thank you for your suggestion. I am really ashamed if my poor English writing level has caused any inconvenience for your reading. I have tried my best to weigh and modify the wording of the paper again and I really hope that I have made some progress. If there is still any inappropriate wording, please do not hesitate to let me know. Thank you again for your patience and kind advice.

Reviewer 2

Comments: The paper "Improving Anti-infrared Radiation" is interesting and actual. I may recommend this paper for publication. I have not any important comments or remarks.

Respond: Thank you for your careful reading and I am really honored for your recommendation.

Thank you for reconsidering our manuscript.

Sincerely,

Furong Liang

Shanghai University of Engineering and Science